# Monoclonal Antibody Therapy for COVID-19: A Retrospective Observational Study at a Regional Hospital

Judith Pannier †, Norbert Nass † , Mohamad-Kamal Yaakoub, Florian Michael Maria Stelzner, Susann Veit, Margarita Kalomoiri, Mahdi Yassine and Gerhard Behre *

Dessau Medical Center and Brandenburg Medical School Theodor Fontane, Auenweg 38, 06847 Dessau, Germany
* Correspondence: gerhard.behre@klinikum-dessau.de
† These authors contributed equally to this work.

**Abstract:** Background: Monoclonal antibodies represent one option for treatment of COVID-19 early after infection. Although large clinical trials have been successfully conducted, real world data are needed to obtain a realistic assessment of the assumed effect on hospitalization rates. Methods: For this retrospective, observational study, clinical data were collected in 2021 from outpatients (402) as well as hospitalized patients (350) receiving monoclonal antibodies Bamlanivimab, Casirivimab/Imdevimab or Etesevimab/Bamlanivimab. These data were compared with data from a control group of patients not receiving antibodies because admission to the hospital was too late for this therapy. Results: Both groups showed a comparable spectrum of risk factors. Due to the late hospitalization of control patients, a higher frequency of severe symptoms, such as fever, dyspnea, syncope and lower viral load, were observed. CRP and leukocytes counts were also higher in the untreated group. Most importantly, hospitalization time was significantly shorter and the number of deaths was also lower in the treated group. Conclusions: Apparently, the application of anti-SARS-CoV-2 antibodies reduced the work load of our hospital as shown by the shorter hospitalization time and lower number of COVID-19-related deaths.

**Keywords:** SARS-CoV-2 virus; COVID-19; monoclonal antibody therapy; Bamlanivimab; Casirivimab; Imdevimab; Etesevimab; hospitalization time





## 1. Introduction

The SARS-CoV-2 virus first appeared in 2019 in China [1] and its spreading has caused a worldwide pandemic, still causing thousands of infections daily worldwide [2]. Soon, monoclonal antibodies (mab) targeting the virus spike protein (S-protein) were developed that can block the adhesion of the virus through the interaction with its receptor ACE-2 [3]. Quickly, several mab products have been approved by the FDA and other agencies, firstly under "emergency use authorization" and later, as regular treatment [4]. In Germany, the Federal Ministry of Health made these antibody preparations available early in 2020 for their application to patients in early stages of the disease when an increased risk for severe COVID-19 exists [5].

Initial large prospective studies showed that these monoclonal antibodies are indeed effective for the prevention of an unfavorable outcome of COVID-19, causing a more rapid reduction of virus load as well as a faster decrease of disease symptoms [6–8].

Nevertheless, there is still a need for "real world data" showing the efficiency in local medical centers; thus, the effective prevention of long hospitalization times and severe outcome. Consequently, several such studies have been published in the last years [9–13]. We have recently published a preprint presenting the data of the first 30 patients in 2021 compared to a control group [14]. The major outcome was a significant reduction in hospitalization time; however, we were not able to show a more rapid decline of viral load upon treatment of this cohort. Here, we extended this study to data on our patients of 2021

and January 2022 to obtain further evidence for the reduction of hospitalization time and improved outcome. To our knowledge, this is the first study evaluating the effectiveness of mab therapy against COVID-19 in Germany.

## 2. Materials and Methods

### 2.1. Patients and Treatment

Information on the option of antibody treatment in the Dessau Medical Center were published on our website and other media and by flyers send to the resident practitioners in the Dessau area. Patients were then referred to our "Corona Antibody Ambulance" by the physicians. This "Corona Antibody Ambulance" was established as a separate ward, only attributed to COVID-19 treatment with monoclonal antibodies. Furthermore, hospitalized patients diagnosed for SARS-CoV-2 infection were offered the mab treatment in case the preconditions such as early state of infection (less than 7 days after symptoms onset), RT-PCR-based virus detection and the presence of risk factors, such as age above 50, coronary heart disease, renal insufficiency and others, were met. Patients not eligible for treatment, mostly as the onset of symptoms was more than 7 days ago when appearing in the hospital, were randomly selected and served as a control group. Patients gave consent for the use of their clinical data and an ethical vote was obtained (Ärztekammer Sachsen-Anhalt, Akz 15/22). Patients were recruited between January 2021 and January 2022. Most patients were enrolled between October and December 2021. Details on numbers and mab treatment can be found in Figure S1.

Monoclonal antibodies (Bamlanivimab, Casirivimab/Imdevimab or Etesevimab/Bamlanivimab) were provided by an initiative of the German Federal Ministry of Health. The antibodies were administered according to the guidelines of the manufacturers' as intravenous infusion over 1 h, followed by an observational period of another hour. All the hospitalized patients were treated for COVID-19 following the same guidelines.

### 2.2. Statistical Analysis

Clinical data were retrieved from our clinical information system, pseudonymized and statistically evaluated using SPSS (IBM, vers. 23). Continuous data were analyzed by the Student's *t*-test and dichotomized data were tested for significance using Fisher's exact test. A *p*-value less than 0.1 was considered as a statistical trend and $p < 0.05$ as statistically significant. For comparison of multiple groups, such as different antibodies, ANOVA and Tamhan T2, post-hoc analysis was applied.

## 3. Results

### 3.1. Distribution of Risk Factors between Treated and Untreated Patients

Risk factors and preexisting illnesses hypertension, diabetes, renal insufficiency, obesity, COPD/asthma, medical immunosuppression, heart diseases, hypothyreosis and the presence of either active or inactive cancers were evaluated in mab-treated versus control patients. These factors were essentially equally distributed in both groups (Table S1). However, there were statistical trends ($p < 0.1$) for a higher incidence of active malignoma in males versus females and inactive malignoma seemed more frequent in untreated versus treated women. Heart diseases were slightly higher in mab-untreated patients and hypothyreosis was less frequent in men versus women. Renal insufficiency was more frequent in untreated versus treated males. We also repeated this analysis in the subgroup of hospitalized patients. Again, only modest differences were observed (Table S3) concerning inactive malignoma and medical immunosuppression. However, the incidence of hypothyreosis was significantly lower in male compared to female patients. Similar results were obtained when we restricted the analysis to Casirivimab/Imdevimab treatment (Tables S2 and S4).

### 3.2. Distribution of COVID-19 Symptoms

We next analyzed whether the appearance of symptoms was different between the treated and untreated patients and between the sexes. Again, we performed this analysis for all the patients and then, limited to hospitalized patients as well as patients treated with Casirivimab/Imdevimab (Tables 1 and S1–S4). Here, several significant differences became obvious. Whereas treated patients reported coughing more frequently ($p < 0.01$), untreated patients suffered dyspnea significantly more often ($p < 0.01$). In addition, syncope ($p < 0.01$) and the loss of taste and smell ($p < 0.1$) was more frequent in treated women. In the hospitalized patients, coughing and dyspnea was similarly skewed and fatigue and pain was more often reported by the treated patients. In blood gas analysis, $pO_2$ and oxygen saturation was only increased in treated women. This became more significant in the hospitalized patients, where mab-treated and especially mab-treated women showed higher levels.

**Table 1.** Selected clinical parameters for hospitalized patients treated with Casirivimab/Imdevimab. Missing data were excluded, causing varying numbers for each parameter. Average $\pm$ standard deviation is shown. For metric parameters, significance was determined using Student's *t*-test; cross tables were analysed using Fisher's exact test. [+] $p < 0.1$; [*] $p < 0.05$; [**] $p < 0.01$. Complete data are shown in the Supplementary Tables S1–S4.

| Parameter | All Patients | | |
|---|---|---|---|
| | **All** | **Untreated** | **Treated** |
| Number | 315 | 90 | 225 |
| Age | $67.1 \pm 17.0$ | $66.9 \pm 20.9$ | $67.2 \pm 15.2$ |
| **Symptoms** | | | |
| Coughing (n/y) % yes | 146/156 51.7% | 52/31 37.3% | 94/125 [**] 57.1% |
| Dyspnea (n/y) % yes | 207/95 31.5% | 44/39 44.0% | 163/56 [**] 25.6% |
| Syncope (n/y) % yes | 280/22 7.3% | 73/10 12.0% | 207/12 [+] 5.5% |
| Blood pressure systolic | $127.7 \pm 20.1$ | $124.9 \pm 22.4$ | $128.5 \pm 19.3$ |
| Blood pressure diastolic | $67.6 \pm 12.8$ | $73.4 \pm 11.7$ | $77.5 \pm 13.1$ [*] |
| **Blood gas analysis** | | | |
| $pO_2$ | $10.0 \pm 3.3$ | $9.7 \pm 4.5$ | $10.2 \pm 2.6$ |
| $pCO_2$ | $4.7 \pm 0.8$ | $4.7 \pm 0.8$ | $4.7 \pm 0.8$ |
| $O_2$ saturation % | $93.6 \pm 4.9$ | $92.3 \pm 4.9$ | $94.1 \pm 4.8$ [**] |
| **Clinical chemistry** | | | |
| CRP (mg/L) | $52.0 \pm 57.9$ | $69.7 \pm 69.5$ | $44.4 \pm 50.4$ [**] |
| PCR (Ct) | $24.8 \pm 5.0$ | $26.7 \pm 4.9$ | $24.0 \pm 4.9$ [**] |
| Hospitalization (d) | $10.3 \pm 9.9$ | $14.4 \pm 11.1$ | $8.6 \pm 8.9$ [**] |
| Death (n/y) % yes | 276/33 10.7% | 71/16 205/17 | 205/17 [*] 7.7% |

In the data obtained from clinical chemistry, especially the inflammation marker C-reactive protein (CRP) stood out as it was significantly higher in all the untreated subgroups of patients. The viral load at hospital admission was evaluated by RT-PCR for the viral S-protein gene. The Ct values were higher in the untreated group when compared to

the mab-treated groups (Table 2), indicating a lower viral load. In summary, the untreated patients, who were largely not eligible for mab therapy because of late admission to hospital although exhibiting a lower viral load, had already developed more severe symptoms than the mab-treated patients, who appeared in the ambulance or hospital within the first week after symptom onset.

**Table 2.** ANOVA test results for significant parameters for all patients (A) and hospitalized patients (B). * indicates significance in Tamhane T2 post-hoc test towards untreated group $p < 0.05$, ** indicates $p < 0.01$. CI = Casirivimab/Imdevimab, B = Bamlanivimab, BE = Etesevimab/Bamlanivimab. $^{\$}$ indicates significance (<0.05) between CI and BE group using this test.

| Parameter | Untreated | CI Group | B Group | BE Group | *p* |
|---|---|---|---|---|---|
| A | | | | | |
| Age | $65.6 \pm 21.2$ | $64.1 \pm 16.2$ | $76.2 \pm 11.4$ | $71.0 \pm 14.7$ | 0.037 |
| CRP (mg/L) | $69.2 \pm 69.3$ | $34.3 \pm 48.2$ * | $10.6 \pm 14.0$ * | $40.0 \pm 39.9$ * | <0.001 |
| PCR (Ct) | $26.7 \pm 4.9$ | $24.2 \pm 5.3$ * | $25.6 \pm 5.8$ | $24.9 \pm 4.3$ | <0.01 |
| $O_2$ saturation (%) | $92.5 \pm 4.9$ | $94.3 \pm 5.4$ * | $92.2 \pm 5.8$ | $94.3 \pm 3.0$ | 0.017 |
| Blood pressure diastolic | $73.6 \pm 11.7$ | $78.4 \pm 12.3$ * | $76.0 \pm 15.2$ | $78.4 \pm 12.8$ | 0.034 |
| Hospitalization time | $14.4 \pm 11.1$ | $4.0 \pm 7.4$ * | $14.0 \pm 12.9$ | $8.8 \pm 9.1$ *$^{\$}$ | <0.001 |
| B | | | | | |
| Age | $65.6 \pm 21.2$ | $67.2 \pm 15.2$ | $76.2 \pm 11.5$ | $72.6 \pm 14.1$ | 0.24 |
| CRP (mg/L) | $69.2 \pm 69.3$ | $44.4 \pm 50.4$ | $10.6 \pm 14.0$ | $41.8 \pm 41.6$ | <0.01 |
| PCR (Ct) | $26.7 \pm 4.9$ | $24.0 \pm 4.9$ ** | $25.6 \pm 5.8$ | $24.7 \pm 4.3$ | <0.01 |
| $O_2$ saturation (%) | $92.5 \pm 4.9$ | $94.1 \pm 4.8$ * | $92.2 \pm 5.8$ | $93.5 \pm 3.0$ | <0.05 |
| Blood pressure diastolic | $73.6 \pm 11.7$ | $77.5 \pm 13.1$ | $76.0 \pm 15.2$ | $79.0 \pm 13.8$ | 0.13 |
| Hospitalization time | $14.4 \pm 11.1$ | $8.6 \pm 8.8$ ** | $14.0 \pm 12.9$ | $11.5 \pm 8.8$ | <0.001 |

### 3.3. Hospitalization Time and Outcome

The most important question for our clinic, however, was whether mab therapy has reduced the number of COVID-19 patients staying in hospital. We first performed this analysis for all the patients and then, separately for Casirivimab/Imdevimab. Additionally, the three monoclonal antibody preparations used in this study were compared with respect to selected parameters (Table 2). First of all, only a few of the outpatients returned to hospital after receiving the antibodies. For all the patients, the overall time of hospitalization was reduced in the mab-treated group from 14.4 to 4.5 days (Tables 1 and 2). This was also significant for hospitalized patients, where this time was reduced from 14.4 to 9.1 days. Interestingly, this was less significant for men. The number of deaths related to COVID-19 was also reduced from 18.1% to 4.1% and from 18.4% to 7.0% in hospitalized patients. The latter was more significant in men. For patients treated with Casirivimab/Imdevimab, the hospitalization time was reduced to 4.0 days (Table S2). For hospitalized patients, this time was reduced to 8.6 days (Table S4). When comparing the three mab preparations, hospitalization time was significantly reduced with Casirivimab/Imdevimab and Etesevimab/Bamlanivimab. For Bamlanivimab, due to the limited number of patients, this effect could not be seen. In all the patients, Casirivimab/Imdevimab seems more effective than Etesevimab/Bamlanivimab in reducing hospitalization time. For hospitalized patients, this effect was lower and not statistically significant (Table 2).

### 4. Discussion

Large prospective studies performed for achieving the approval of mabs for the treatment of early COVID-19 have demonstrated their positive effect on outcome and symptoms for patients [15]. Several mabs can now be applied for the therapy of patients

with a higher risk for severe COVID-19 disease [4]. Important risk factors are hypertension, obesity, chronic respiratory and renal diseases as well as age above 50 years. With the ongoing evolution of the SARS-CoV-2, the specificity of mabs for mutated S-protein genes became increasingly important [16]. Initially, single mabs were used for treatment, then followed by the application of two mabs with different target epitopes to circumvent the development of resistance by a mutation in the epitope [8]. Alternatively, highly conserved parts of the S protein were chosen as antibody targets. In our study, we followed the recommendations for the use of mabs according to the prevalent virus variant and the availability of the antibody preparations. As result, Bamlanivimab was the first mab in use, followed by Casirivimab/Imdevimab for most patients as the delta variant was dominating, especially from October 2021 to January 2022. Bamlanivimab/Etesevimab was only shortly in use (September to November 2022) parallel to Casirivimab/Imdevimab. In our data analysis, we focused on the main mab preparation Casirivimab/Imdevimab, which was applied in 81% of all cases during a significant surge of the delta variant (Figure S1). Nevertheless, we cannot exclude that some patients received a mab that did not match to the virus variant present.

Our main interest in this study was obtaining an assessment whether the application of mabs did, indeed, reduce the COVID-19-related work load of our hospital. Altogether, our data suggest that this was, indeed, the case. Mab-treated patients showed a shorter hospitalization time and a lower number of COVID-19-related deaths. However, when comparing the mab-treated- with the untreated group, we observed that the untreated patients exhibited more severe symptoms, especially concerning more frequently reported dyspnea, higher inflammation as deduced from CRP-levels, but a lower viral load as shown by the RT-PCRs' Ct values. We propose that this is due to the fact that these patients are mainly admitted to our hospital late after symptom onset, and repeatedly by the emergency service after collapsing. Thus, these patients did not qualify for mab treatment. The finding that the untreated patients showed higher Ct values in RT-PCR can be caused by the longer infection time, which resulted in an already declining viral load in nasopharyngeal swabs [17]. Another control group was not available as it would be unethical to withhold a proven treatment from eligible patients. However, the distribution of risk factors for a severe COVID-19 outcome was comparable to the treated group. Nevertheless, it cannot be excluded that the apparently more severe disease in our control group contributed to the longer hospitalization time and increased number of deaths. We think this suggests that the application of the anti-viral antibodies early after infection is essential for therapy success. When comparing the outcome of Casirivimab/Imdevimab and Bamlanivimab/Etesevimab, Casirivimab/Imdevimab seems more effective. These data might reflect that Bamlanivimab/Etesevimab was reported to be less effective against the delta variant of the SARS-CoV-2 virus [18]. However, the number of patients treated with these mabs was also low causing decreased statistical power.

We additionally observed differences between male and female patients concerning pre-existing illnesses, especially for hypothyreosis, which was consistent with data on the distribution of this preexisting illness in general. Moreover, that the male sex can be considered as a risk factor [19] is already known and, therefore, not unexpected.

## 5. Conclusions

Although the control group was not perfectly comparable to the mab-treated group, our data suggest that mab treatment reduced the COVID-19 related work load of our hospital.

**Supplementary Materials:** The following supporting information can be downloaded at: https://www.mdpi.com/article/10.3390/idr15010013/s1. Figure S1: Recruitment of the patients; Table S1: Statistical analysis of clinical parameters for all patients; Table S2: Statistical analysis of clinical parameters for patients treated by Casirivimab/Imdevimab; Table S3: Statistical analysis of clinical parameters for all hospitalized patients; Table S4: Statistical analysis of clinical parameters for all hospitalized patients treated with Casirivimab/Imdevimab.

**Author Contributions:** Conceptualization, J.P., N.N. and G.B.; methodology, J.P. and N.N.; validation, J.P., N.N. and G.B.; investigation, J.P., N.N., M.-K.Y., F.M.M.S., S.V., M.Y. and M.K.; resources, J.P. and G.B.; data curation, N.N.; writing—original draft preparation, N.N.; writing—review and editing, J.P. and G.B.; supervision, G.B.; project administration, G.B.; funding acquisition, G.B. All authors have read and agreed to the published version of the manuscript.

**Funding:** This research received no external funding.

**Institutional Review Board Statement:** The study was conducted in accordance with the Declaration of Helsinki, and approved by the Ethics Committee of the Ärztekammer Sachsen-Anhalt, Akz 15/22 (26 April 2022).

**Informed Consent Statement:** Informed consent was obtained from all the subjects involved in the study.

**Data Availability Statement:** Detailed data are available from the authors upon reasonable request.

**Acknowledgments:** The authors wish to thank all the staff of the clinic for internal medicine I for their continuous support.

**Conflicts of Interest:** The authors declare no conflict of interest.

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
