# Peer review of "Monoclonal Antibody Therapy for COVID-19: A Retrospective Observational Study at a Regional Hospital"

_2036-7449, doi:10.3390/idr15010013_

Round 1

Reviewer 1 Report

Pannier et al describes an observational study of the treatment of Covid-19 by monoclonal antibodies in a German hospital. The number of treated patients is with around 650 quite significant, the control group of ~100 patient is smaller. Several parameters were measured like symptoms, pre-existing illness, blood gas analysis and clinical chemistry and evaluated under different aspects like hospitalization and gender. The main conclusion is the reduction of hospitalization time and thereby reduction in work load of the hospital staff.

In general, this is a very interesting observational study including a large number of treated patients. Yet, the control group is not perfect as this group probably had a more severe symptoms (collapsing as reason to be admitted to the hospital) as it was mainly excluded by a late entry into the hospital. This is also stated in the discussion by the authors themselves. In my opinion that weakens the conclusion of a reduction from 14 to 9 days immensely. Nevertheless, I totally follow the reason that not treating patients by mAbs if possible is unethical. Yet, I am wondering if there are not better control groups available maybe in other hospitals were mAbs were not given as the standard treatment.

Even if many parameters were measured, I am missing the viral load. Especially after it was mentioned in the introduction (Line 48-49) that a reduction could not be observed in a smaller cohort before.

One major point is that the data are not discriminated by the different mAbs used, as it is known that mAbs differ in neutralization capacity. If numbers were too low for Bamlanivimab and Casirivimab/Imdevimab I suggest to exclude them totally from this analysis. At least I would like to see a comparison of the data of only Etesevimab/Bamlanivimab treatment compared to Bamlanivimab or Casirivimab/Imdevimab or Etesevimab/Bamlanivimab treatment. Otherwise, I expect that the authors analysed this and that they can state that they could not observe significant differences for the different treatments.

Following this, I am also wondering if the patients have been sequenced for the virus subtype to validate that the mAb treatment could be helpful at all. I guess that one could not wait for sequence results before application of the mAb, but afterwards it would be helpful to exclude data of patients with the “wrong” subtype for the antibody (these patients might be a suitable control group) and probably even increase the measured positive effect of an mAb treatment. If this data is not available the general ratio of the “wrong” subtypes during the overall period of the study (e.g. 3 % of beta which is not recognized by Etesevimab/Bamlanivimab) should be added.

For a better overview I would also suggest to shorten the tables and only list the parameters that were significant (or showed a trend) in at least one of the tables. The tables containing all data should be moved to the supplement.

Still, all over it is a very interesting study for the scientific community containing a lot of data and after some revision definitely publishable.

Author Response

Please refer to the attached file. I have prepared a response letter for the referees.

Reviewer 2 Report

The authors administered SARS-CoV-2 monoclonal antibodies to COVID-19 patients and followed up. Although many clinical parameters were similar between treated and untreated patients, some important differences were revealed. Especially, length of hospitalization and mortality were significantly reduced with mab-treatment. These data are considered to be important findings that mab-treatment can contribute not only to patient recovery but also to the reduction of the work load of medical staff involved in patient treatment.

I have same comments.

1.  It is necessary to indicate when these data were collected.

2.  Were there any relationships between the amount of virus detected by RT-PCR and hospitalization or mortality in this study?  (Virus load was not described in this manuscript.)

3.  P14, lane 147 to 149

I think it would be better to indicate the approximate time period for each. For example, how about a description from January to March or from the early to the late middle term?

4.  P2, lane 57

What is 'Corona antibody ambulance'?

5.  P2, lane 58

Does 'Corona' mean SARS-CoV-2?

6.  P8, lane 125

Although it is described as '7.1%' in the text, it is described as '7.0%' in Table 2. This value should be correct to the correct value.

Author Response

(The authors gave the same response as above.)

Reviewer 3 Report

OVERALL

The manuscript entitled: "Monoclonal Antibody Therapy for Covid-19: A Retrospective 2 Observational Study at a Regional Hospital " is a communication by Pannier et al. about a retrospective study in a German Regional Hospital where monoclonal antibody (Bamlanivimab, Casirivimab/Imdevimab or Etesivimab /Bamlanivimab) were used as therapy for outpatients or hospitalized patients diagnosed with COVID-19.

MAJOR COMMENTS

I respectfully disagree with the authors when they stated that “A mab-specific analysis was therefore not promising due to low numbers” (line 152-153) because in the previous lines (151 and 152) they explained that in 81% of all the cases during the delta variant outbreak, the cocktail Casirivimab/Imdevimab was used as therapy. There are substantial molecular changes among the SARS-CoV-2 variants that might influence clinical findings, therefore, for me, it would be more relevant to present 81% of the population of the study and focus the results on the Delta variant. Alternatively, the results could be presented with all the cases, establishing the specific antibody that was used, even though other variants did not reach a statistical difference but probably a trend might have been seen, this could strengthen your discussion including viral variant, therapy, and clinical findings. In any case, the clinical data had to be regrouped and reanalyzed.

MINOR COMMENTS

I guess that the authors meant variant instead subtype (lines 147 and 151).

There are several typos, some of which are due to a wrong spelling (innapetenz instead of inappetence); lack of uniformity of the use of British English vs American English (Hospitalisation vs Hospitalization); misspelling of COVID-19 (in some parts of the manuscript was written as Covid-19 (title), Covid19 (lines 100, 120) or Covid 19 disease (line 139) this last form is incorrect since the acronym includes the word “disease”; extra whitespaces between words pressure systolic and pressure diastolic; an extra quotation mark at the end of the sentence (line 179); homogenize the word “table”, sometimes is written at the beginning with a capital letter and other with a small letter.

I could not find adipositas in the dictionary, I assume that the authors meant obesity, please clarify or change.

I guess the best word that fits the context expressed in the paragraph from lines 165-168 is sex instead of gender.

Author Response

(The authors gave the same response as above.)

Round 2

Reviewer 3 Report

I am glad to see the newest version, it is clearer results section (the new Figure 1 and the distribution in supplementary tables) and more insightful discussion. There are still some typos that easily can be fixed.

-Lines 77-82, the text should not been bolded.

-Line 89, a comma is missinf after "hypothyreosis".

-Line 147, there is an extra period after "seen".

-Lines 170 and 179, change subtype for variant.

-In all tables say "hospitalisation" instead of "hospitalization".

Author Response

Thank you very much for your encouraging remarks. We have made the changers requested as described in the reply

Reviewer comments

Referee 3

I am glad to see the newest version, it is clearer results section (the new Figure 1 and the distribution in supplementary tables) and more insightful discussion. There are still some typos that easily can be fixed.

-Lines 77-82, the text should not been bolded.

Sorry, I am not sure how this happened… done.

-Line 89, a comma is missinf after "hypothyreosis".

With all respect, I cannot find this missing comma.

-Line 147, there is an extra period after "seen".

This has been removed removed.

-Lines 170 and 179, change subtype for variant.

Sorry for the mistake. “Subtype” has been exchanged with “variant” throughout the manuscript.

-In all tables say "hospitalisation" instead of "hospitalization".

Hospitalisation (British English) was replaced by hospitalization (American English) in all 4 supplementary tables.